# Postural Control Measurements to Predict Future Motor Impairment in Preterm Infants: A Systematic Review

**DOI:** 10.3390/diagnostics13223473

**Published:** 2023-11-18

**Authors:** Jennifer Bosserman, Sonia Kelkar, Kristen D. LeBlond, Jessica Cassidy, Dana B. McCarty

**Affiliations:** 1Physical Medicine and Rehabilitation, Johns Hopkins Hospital, Baltimore, MD 21205, USA; jbosser3@jh.edu; 2Department of Health Sciences, University of North Carolina at Chapel Hill School of Medicine, Chapel Hill, NC 27599, USA; 3Physical Therapy and Occupational Therapy, Duke Health, Durham, NC 27705, USA; 4North Carolina Children’s Hospital, Chapel Hill, NC 27599, USA

**Keywords:** postural control, center of pressure, preterm infant, force plate, postural measurement

## Abstract

Preterm infants are more likely to demonstrate developmental delays than fullterm infants. Postural measurement tools may be effective in measuring the center of pressure (COP) and asymmetry, as well as predicting future motor impairment. The objective of this systematic review was to evaluate existing evidence regarding use of pressure mats or force plates for measuring COP and asymmetry in preterm infants, to determine how measures differ between preterm and fullterm infants and if these tools appropriately predict future motor impairment. The consulted databases included PubMed, Embase, Scopus, and CINAHL. The quality of the literature and the risk of bias were assessed utilizing the ROB2: revised Cochrane risk-of bias tool. Nine manuscripts met the criteria for review. The postural control tools included were FSA UltraThin seat mat, Conformat Pressure-Sensitive mat, Play and Neuro-Developmental Assessment, and standard force plates. Studies demonstrated that all tools were capable of COP assessment in preterm infants and support the association between the observation of reduced postural complexity prior to the observation of midline head control as an indicator of future motor delay. Postural measurement tools provide quick and objective measures of postural control and asymmetry. Based on the degree of impairment, these tools may provide an alternative to standardized assessments that may be taxing to the preterm infant, inaccessible to therapists, or not sensitive enough to capture motor delays.

## 1. Introduction

There is a risk of motor impairment in all preterm infants born <37 weeks of gestation; however, the risk is highest in infants born moderately preterm (32–34 weeks of gestation) at 20.6% and very preterm (<32 weeks of gestation) at 36.1% [1,2]. When comparing fullterm and preterm infants, the risk of motor impairment ranges from 2 to 7% compared to 54–64%, respectively [3]. There are notable differences between the movement patterns of preterm and fullterm infants. Preterm infants demonstrate a lower quality of spontaneous movements, with descriptions such as low fluency, less variety, and impaired sequencing [4]. Preterm infants are also more likely to display abnormal or absent fidgety movements, ref. [5,6] which is a highly sensitive indicator of future motor impairment at 12 weeks of age. Preterm infants also lack postural complexity, defined as the use of a variety of postural control strategies, as compared to healthy term infants [7].

Preterm infants are more likely to display body and head asymmetry and show preference for extension patterns than fullterm infants [6]. These asymmetrical patterns may be attributed to the development of increased power in the extensor muscle groups in preterm infants. Increased muscle power [8] results in the hyperextended posture commonly observed in preterm infants. This posture further leads to difficulties in maintaining midline orientation [8].

The neonatal intensive care unit (NICU) environment is not optimal for neonatal neuromotor development for a variety of reasons, including noise levels, lighting, and other noxious stimuli, as well as suboptimal musculoskeletal support [9]. Preterm infants are vulnerable to the effects of gravity on alignment, posture, mobility, respiratory abilities, and the shaping of the musculoskeletal system [10,11]. Without the intrauterine environment facilitating a flexed posture and limiting extremity movement, preterm infants succumb to the weight of gravity and begin to favor an extended posture. In an attempt to gain postural stability in the absence of uterine wall restraint [10,11], preterm infants often extend their trunk and extremities further into the flat surface they are placed upon, resulting in commonly described postures of exaggerated cervical lordosis and hyperextension [10]. This combination of the effects of gravity and hyperextended posture leads to weak and overstretched muscles of the anterior neck and trunk, interferes with purposeful self-soothing movements directed towards the midline, and contributes to developmental delay [12].

Extended posture and associated asymmetries in movement can result in head or positional preferences. The prevalence of such positional asymmetries ranges from 45 to 79% of preterm infants [13,14]. Head turn preferences in preterm infants are associated with sub-optimal reflexes, decreased maturation of gross motor movements, and the development of torticollis and deformational plagiocephaly during infancy [13,14]. These impairments, if not fully addressed, further contribute to delays with increasing infant age, including impaired fine motor skills, asymmetrical gait patterns, and postural asymmetries [13,15].

Moderate to severe neuromotor and sensory disabilities are highly prevalent in extremely and very preterm infants born between 22 and 34 weeks of gestation [16], requiring early assessment and intervention. Preterm infants born 24–31 weeks of gestation remain in the NICU for a range of 34–123 days [17]. Due to increased risk of motor delay, these infants often receive physical and/or occupational therapy services during hospitalization. Current evidence supports parent- and therapist-delivered motor intervention to improve motor and cognitive developmental outcomes in preterm infants [18], and immediate, ongoing therapy services after hospital discharge to reduce the risk of developmental delays; however, there is often a delay in the initiation of therapy services after NICU discharge [19,20], especially if no significant motor impairment or diagnosis has been documented using a standardized or objective measure. Standardized assessments vary based on the appropriate age for administration, domains of function tested (e.g., motor, neurobehavior), applications, and predictive validity. Few standardized assessments are sensitive enough to detect developmental delays for infants at or near term-equivalent age [21,22], which is often the age of the infant at the time of NICU discharge.

Standardized infant assessments also vary greatly by administration requirements, with many necessitating costly training programs to learn and administer testing [23]. While these training programs are in place for the essential purpose of ensuring reliable and valid results for clinical and research applications, the rigorous requirements are often out of reach for therapists and the NICUs they serve due to a lack of continuing education funds and travel requirements. In the absence of extended time for clinicians to attend training programs and funding to pay for such programs, it is prudent to identify objective measures that indicate potential delay that can be used, assessed, and understood by a variety of clinicians and researchers. Quantitative measurements such as center of pressure (COP) and variability of movement have been shown to be predictive of motor impairment or delay in preterm infants [24], but these measures are not currently used in the clinical setting to identify infants at risk for movement delay.

Pediatric therapist researchers are advocating for the expanded use of technology in clinical settings to detect early motor delay [25]. Postural measurement tools, including portable pressure-sensitive mats and force plates, may be effective in measuring COP in preterm infants, and therefore useful for detecting early delays in high-risk infant populations [24]. Additionally, the use of wearable sensors, including inclusive clothing, exoskeletons, and smart tracking devices, are being examined in high-risk infant populations to determine potential ways that this technology can assist with understanding how specific movement characteristics may enhance or detract from the infant’s developmental trajectory [25].

The use of such technologies expands opportunities for the use of artificial intelligence to assist in the early and accurate diagnosis of neuromotor disabilities. Gaining a better understanding of the role of early postural control deviations as measured by force plates or sensors allows researchers to use these technologies to build algorithms that detect and quantify movements associated with future motor impairments [26].

The purpose of this review was to evaluate the existing evidence regarding the use of technology, specifically, pressure mats or force plates, to measure both linear and non-linear measures of postural control and movement in preterm infants. We also evaluated how those measures differ in preterm and fullterm infants and how these differences may predict future motor impairment or disability in this population.

## 2. Methods

The following inclusion criteria were used to select studies: (1) articles include infants born at or prior to 37 weeks, (2) measurements were collected in supine, (3) measurements were collected using a pressurized mat or mattresses, and (4) articles were in reference to humans. Exclusion criteria included the following: (1) a study population of infants born after 37 weeks of gestation, (2) measurements taken in positions other than supine, or (3) articles with reference to animals. We did not include gray literature or dissertations.

The protocol for the review was drafted using the Preferred Reporting Items for Systematic Reviews and Meta-analysis Extension (PRISMA) [27] and was registered to the Open Science Framework (OSF, registration DOI number: 10.17605/OSF.IO/G82WK) [28]. The objective of this systematic review was to answer the question: “In preterm infants, can center of pressure (COP) measurements and variability of movement measurements in supine help determine the risk of motor delay in infancy?”. A search strategy using keywords was developed by the primary author (JB) in consultation with a university librarian and included “(“Infant, Newborn” [MeSH] OR “Premature Birth” [MeSH] OR Neonatal [tiab]) AND (“Postural Balance” [MeSH] OR “Pressure, Mat*” [tiab] OR “Multisensor” [tiab] OR “Force Plate” [tiab])”. Four databases were searched in September 2023 (PubMed, Embase, Scopus, and CINAHL). One investigator (JB) used MeSH headings and text words to complete the search. Results were imported to Covidence [29], a systematic review production tool for title/abstract/full-text review and data abstraction.

Two reviewers (JB, SK) independently reviewed and extracted papers that met the inclusion criteria for full text review through methods consistent with the PRISMA guidelines [30]. Any disagreement about inclusion was discussed amongst the reviewers (JB, SK), and the senior author (DM) made the final determination. Papers that passed the full-text review were evaluated with an extraction table based on recommendations from the Cochrane Collaboration [31] and included the following characteristics: study aims, study design, data sources, study population, outcome measures, data analysis strategy, postural measurement tool, results, implications, strengths, and limitations. Data extracted were then reviewed using a descriptive approach to summarize key findings.

The quality of the literature and risk of bias were rated utilizing the ROB2, revised Cochrane risk-of-bias tool, for randomized trials [32] for each included study. Independent assessments were completed by two reviewers (JB and DM), and full agreement was reached after discussion.

## 3. Results

### 3.1. Study Selection

The initial keyword search identified nine hundred and one studies. Two hundred and fifty-three of these were excluded as duplicate studies from multiple databases. The remaining six hundred and forty-eight studies underwent title and abstract screening based on the inclusion and exclusion criteria. A full-text review for eligibility was completed for twenty-six full-text studies, and nine met all eligibility criteria (Figure 1).

### 3.2. Characteristics of the Included Literature

Of the included studies, six were prospective cohort studies [7,24,33,34,35,36], two were cross-sectional studies [37,38], and one was a case study [39]. The characteristics of these studies are noted in Table 1. All studies were conducted in the United States [7,24,33,37,38,39], Norway [34,35], or Poland [34].

### 3.3. Participants

All studies included infants born preterm (<37 weeks of gestation). The majority of studies (*n* = 6) also included a control group of fullterm infants with typical motor control (Table 1) [7,24,33,34,35]. Other participant characteristics reported were variable and are included below in the results section.

### 3.4. Quality Assessment

The results of the quality assessment can be seen in Table 2. Due to the nature of the infant population and study designs, blinding and random allocation did not occur. This resulted in all nine studies receiving the rating of “high concern” for Domain 1, risk of bias associated with the randomization process, as well as Overall Risk of Bias, per the scoring criteria [32]. Two studies [34,36] also received “high concern” in other domains due to deviation from the intended intervention.

### 3.5. Postural Control Measurement Systems and Measures of Postural Control

Postural control tools and measurement parameters varied between studies as seen in Table 1. Two studies utilized the FSA UltraThin seat mat (Vista Medical Ltd., Manitoba, MB, Canada) [37,38] to measure the maximum pressure value, the ratio of head and pelvis to trunk pressure, and COP. The Conformat Pressure-Sensitive mat (Tekscan Inc., Norwood, MA, USA) was utilized in three studies [7,33,39] to measure COP, the magnitude and complexity of movement, head control, and reaching ability. The Play and Neuro-Developmental Assessment (PANDA) gym (Penn Center for Innovation, Philadelphia, PA, USA) [24] was used to measure limb and trunk kinematics and COP measurements. Lastly, three studies used standard force plates by AMTI (Advanced Medical Technologies Inc, Watertown, MA or Kistler (Kistler Instrument Corp., Amherst, NY, USA) [34,35,36] to measure postural adjustments with reaching and COP displacement.

Both linear and non-linear measures of postural control were used in the reviewed studies. Linear measures such as path length quantify the amount of COP variability [40]. Generally, in the adult population, high variability in COP is interpreted as postural instability, whereas lower COP variability indicates greater postural control; however, the studies reviewed for this manuscript noted greater COP variability in healthy fullterm control infants as compared to preterm infants, who demonstrated less complexity in movement [24].

Non-linear metrics, which incorporate time into COP variability, were also used for postural control analysis. These non-linear metrics quantify the amount of randomness, fluctuation, and unpredictability during dynamic movement [40]. As observed in linear postural metrics, preterm infants actually demonstrated smaller amounts of entropy, or randomness, than fullterm infants, indicating less variability of complex movement [38].

#### 3.5.1. FSA UltraThin Seat Mat

The Force Sensing Array (FSA) UltraThin seat mat is a pressure-sensitive mat that is commonly used in wheelchair seating systems. The FSA seat mat includes a 4D pressure mapping system [41] that measures the total duration of trunk flexion, extension, or neutral positioning, determined according to the total number of frames the infant’s trunk was in for each position and multiplying the total consecutive frames by the sampling period (200 ms) [37]; approximate entropy, a ratio that estimates the randomness, fluctuation, and unpredictability of time-series data [39]; and root mean square values, the standard deviation of the displacement of the COP in the caudal–cephalic and medial–lateral directions [42,43].

In a cohort of 33 infants aged 38.30–42.30 weeks corrected age, researchers found that term infants (mean gestational age of 38.9 weeks) spent significantly 81% of the awake segment in flexion or neutral (*p* = 0.027), and only 74.91% of preterm infants (mean gestational age of 31.9 weeks) spent the awake segment in flexion or neutral (*p* = 1.52) [37]. In a cohort of 32 infants at 41–43 weeks corrected age, preterm infants exhibited larger root mean square values (preterm = 1.11 cm, term = 0.83 cm; *p* = 0.01) and smaller approximate entropy values (preterm = 1.11, term = 1.19; *p* = 0.02) in the caudal–cephalic direction than term infants [38]. Authors concluded that smaller approximate entropy and larger root mean square values in preterm infants suggest less complex, repetitive movement, and less stable posture in the caudal–cephalic direction [38].

#### 3.5.2. Conformat Pressure-Sensitive Mat

A Conformat Pressure-Sensitive mat is a portable and lightweight seating and positioning system often used for wheelchair pressure mapping, which provides information on pressure distribution and the center of force trajectory [7]. Dusing et al. used the Conformat Pressure-Sensitive mat to measure the root mean square and approximate entropy values (as defined in the previous section) in the caudal–cephalic and medial–lateral directions [7,33,39].

Results from a cohort of three infants 2–6 months old demonstrated an interaction between condition and age, in the caudal–cephalic direction of postural variability (*p* = 0.03), and that preterm infants demonstrated low complexity movements in the caudal–cephalic direction, ref. [39] indicating that decreased postural complexity before the development of midline head control may be an indicator of future motor delay.

#### 3.5.3. The PANDA Gym

The Play And NeuroDevelopmental Assessment (PANDA) includes an array of toys with sensors in them, a camera-based computer vision system, and a mat structure covered in carbon fiber [24]. This gym also includes a PVC pipe above the platform for toy suspension and to support the video system [24]. In a cohort of 15 infants aged 3–11 months old, the PANDA gym measured seven variables including path length, the total distance an object moved from its initial position to its final position; ExcursionX/Y, with ExcursionX being the farthest distance in the medial–lateral direction, or side-to-side shifting, and ExcursionY being the farthest distance in the caudal–cephalic direction, or vertical shifting; and ElipseArea, the scatter of COP in the X and Y directions [24].

Vertical displacement (ExcursionY) was significantly lower in the preterm group compared to the term group (difference = 3.65 cm, 95% confidence interval (CI): 0.13–7.17 cm, *p* = 0.043), demonstrating a smaller distance traveled in the caudal–cephalic direction, with minimal vertical shifting [24]. The COP variability (EllipseArea) was significantly lower in the preterm vs. term group (difference = 2.3 cm, 95% CI: 1.06–4.84 cm, *p* = 0.038). These results indicate less movement variability in preterm infants, specifically in the caudal–cephalic direction. The total distance traveled (path length) was significantly higher in the preterm group compared to the fullterm group for three conditions (no toy 153.4 vs. 101.3 cm, *p* = 0.0054; bilateral reach 146.1 vs. 87.5 cm, *p* = 0.0088; and unilateral reach 176.6 vs. 112.2 cm, *p* = 0.0005), demonstrating increased movement from the initial position to the final position in preterm infants. Lastly, in the group that was identified as having impaired motor control at 2 years of age, as determined from a medical record review, path length was found to be higher in all conditions (no toy 155.6 vs. 115.9 cm, *p* = 0.033; bilateral reach 158.4 vs. 100.1 cm, *p* = 0.003, and unilateral reach 223.1 vs. 122.2 cm, *p* < 0.0001) [24].

#### 3.5.4. Force Plates

A multi-axis force plate is capable of measuring all dynamic motion sequences, including abruptly changing forces [40]. In two studies [32,33], total body COP was analyzed from force plates using several parameters, including the path length (as defined above), the length and duration of the movement path/time, the number of directional changes in COP displacement, and the maximum velocity (Vmax): the maximum speed in which the infant moves in the cranial–caudal and medial–lateral directions [34,35].

Findings from a long-term follow-up study conducted by Fallang et al. [35] showed that in fifty-two 4-month-old infants, a lower maximum velocity of COP and smaller displacement of COP in the medial–lateral direction were related to coordination problems at 6 years of age (*p* = 0.04). At 4 and 6 months, performance below the 15th percentile on the Movement ABC at 6 years was associated with a lower Vmax of COP in the medial–lateral direction at 4 months (*p* = 0.02) and 6 months (*p* = 0.03) and a lower number of cranial–caudal oscillations (4 months *p* = 0.02, 6 months *p* = 0.01) [35]. In a related study, Fallang et al. found that the total body COP in preterm infants differed from fullterm infants due to a smaller COP distance travelled during reaching in both the cranial–caudal and medial–lateral directions, demonstrating relatively immobile postural behavior [34].

A recent study by Kniaziew-Gomoluch et al. [36] used force plates to examine postural control in 37 preterm infants born between 24 and 33 weeks of gestation at 12–14 weeks corrected age. Infants simultaneously were video-recorded for the General Movements Assessment. Researchers found significant differences in all parameters of spontaneous COP displacement between infants with normal fidgety movements and those without fidgety movements (*p* < 0.05). Using the Intraclass Correlation Coefficient for test–retest data, all parameters measured in supine were considered to have moderate to good reliability [36].

## 4. Discussion

While there is still much to learn about the quantitative measurement of postural control in preterm infants, the available evidence demonstrates that tools such as force plates and pressure mats are feasible for the measurement of infant postural control and asymmetry. Further, these tools identified differences in preterm infant movement as compared to fullterm infant movement. Quantitative measurements of trunk positioning during spontaneous activity may be a reasonable and useful measure to identify infants at high risk for motor impairment or disability and those who are not [24,36,37]. In opposition to how postural complexity is interpreted in adults, several studies indicate that reduced postural complexity in infants before development of midline head control may indicate future motor delay [7,33,37,38]. Evidence also suggests that these postural control measures are sensitive to the later development of neuromotor dysfunction. One study found associations between postural control parameters at 4 and 6 months and motor scores at 6 years of age [35], and another study found that postural control parameters were significantly different between groups of preterm infants with normal and abnormal fidgety movements—an early predictor of cerebral palsy [36].

Postural measurements have been used most consistently in research applications, but the results from this systematic review demonstrate potential for clinical application to support early identification of infants with motor delay—potentially as early as term-equivalent age. Specific atypical measurements of postural control that have been associated with future motor impairment include COP path length, COP extent, variety of movement, and speed of movement, especially in the caudal–cephalic direction [24,38]. Sensitive measures that predict future motor impairment and characterize some preterm infant movement characteristics include predictable and repetitive COP movement in the caudal–cephalic direction, a relatively immobile posture, a lack of successful reaching by 4 months, and inadequate reaching quality at 6 months in supine [7,34,36].

Currently, many preterm infants do not qualify for early intervention services when assessed based on state-by-state qualification standards for Part C of the Individuals with Disabilities Education Improvement Act [44]. Additionally, the available standardized assessments may not be sensitive enough to capture the extent of an infant’s delay prior to NICU discharge. Because standardized assessments generally quantify the infant’s capacity to exert postural control in specific developmental positions, but do not always quantify the quality and characteristics of the infant’s movement, subtle postural differences may be missed [21]. For example, in a recently published study, Wang et al. discusses that the General Movement Assessment (GMA) can identify absent or abnormal fidgety movements and has 98% sensitivity for the diagnosis of cerebral palsy at 12 weeks [36], but a limitation of this assessment is the absence of a measurement tool used to quantify these movements [21]. The Motor Optimality Score (MOS), a detailed scoring of the GMA, is currently in the early stages of reliability testing [45] but requires advanced GMA certification to administer. Based on the recent findings of Kniaziew-Gomoluch et al. [36], COP parameters as measured by force plates are sensitive enough to detect differences between infants who demonstrate normal fidgety and abnormal fidgety movements.

Of the measurement tools described in this review, perhaps the most promising for future clinical use is the PANDA gym [24]. With its portable design and ongoing research using machine learning to develop algorithms to produce measurements and relevant scores, the PANDA gym has the potential for widespread use in various clinical settings to diagnose early movement dysfunction. While early validity assessments of this mat system are promising, additional reliability, validity, and sensitivity to change testing should be conducted prior to clinical applications. Additionally, findings from Kniaziew-Gomoluck et al. [36], demonstrate a correlation between force plate-measured postural control parameters and absent fidgety movements between 12–14 weeks post-term, indicating a potential clinical usefulness for early cerebral palsy detection [36,46]. COP path length, which is measured via the PANDA gym and force plates, consistently differed between fullterm and preterm infants in the studies we reviewed and appear to be early indicators of motor delay [36]. Most studies did not address whether force plate and pressure mat technologies can be easily disinfected between uses in the highly vulnerable preterm infant population; however, the carbon fiber core dragon plate used in the PANDA gym can be easily cleaned with soap and water or disinfectant wipes between use [24]. This plate is covered with foam padding or a blanket for infant comfort for single patient use to decrease the spread of infection.

Limitations of this study include the acknowledgement that the use of pressure-sensitive mats and force plates in the clinical setting may not be easily attained due to the high cost of equipment and maintenance requirements; however, with more advanced technology, newer devices are becoming available that may increase affordability and portability necessary for clinical spaces. We also acknowledge that a shift in clinical practice and eligibility standards would be necessary to use atypical postural control measurements to quantify motor delays. Furthermore, clinicians would need additional training to collect and interpret these data in a meaningful and objective way.

This study provides ample evidence for the use of pressure mats and force plates to measure postural control, asymmetry, and variability of movement in preterm infants, but future research is needed to employ this globally. Future research should focus on the validity, predictive ability, sensitivity to change over time, and quantification of severity necessary to detect future motor impairments [24]. Further, infants should be assessed over a shorter time period to improve the test–retest reliability of these methods [33]. A longitudinal follow-up of high-risk infants and those who later develop motor impairment would also be useful in determining which infants can adapt to changing task demands based on postural control in early infancy [4]. Studies presented in this systematic review support the association between the observation of reduced postural complexity prior to the observation of midline head control as an indicator of future motor delay [7,33,37,38]. This observation should be verified utilizing larger sample sizes with a long-term follow-up. Future research is also necessary to determine critical periods of time in which postural complexity has a greater impact on development and optimal variability of movement, as well as which occupational therapy and physical therapy interventions best mitigate the risk of delay [37]. Based on the risk-of-bias assessment for the manuscripts assessed in this study, the blinding of researchers or the separation of tasks for collecting and reducing the data should be considered in the methodology of future studies.

## 5. Conclusions

There is a need to identify impairments in early posture and movement complexity in order to avoid delays in post-NICU therapy services. Altered posture and movement in preterm infants limits the infants’ ability to explore the world around them, perform variable movements, use perceptual information to modify movement, and practice a variety of postural control strategies [33]. Postural measurement tools such as force plates and pressure-sensitive mats provide quick and objective measures of COP and asymmetry, and, based on the degree of impairment in postural control and movement, may indicate future motor impairment, providing an alternative to the application of standardized assessments for the quantification of developmental delay.

## Figures and Tables

**Figure 1 diagnostics-13-03473-f001:**
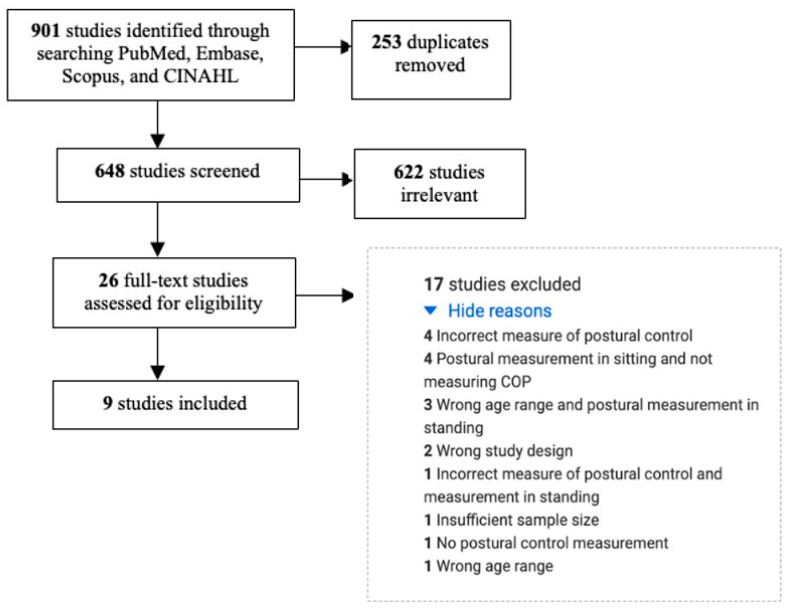
Study selection process.

**Table 1 diagnostics-13-03473-t001:** Study characteristics.

First Author, Year	Aim	Study Design	Study Population	Postural Measurement Tool	Outcome
Dusing et al., 2009 [39]	To determine whether infants born at full term and infants born preterm differ in their COP movement variability characteristics, evaluated both linearly and nonlinearly while positioned supine.	Cross-Sectional Study	47% fullterm	FSA UltraThin Seat Mat	Infants born pre-term exhibited larger root-mean-squared values in the caudal–cephalic direction than infants born full-term.
Dusing et al., 2005 [38]	To compare trunk position in supine of infants born preterm and at term. A secondary purpose was to determine the feasibility of using pressure data to assess trunk position.	Cross-Sectional Study	45% fullterm	FSA UltraThin Seat Mat	Infants born preterm differ in their trunk positions immediately after birth as demonstrated by decreased time spent in flexion or neutral.
Dusing et al., 2016 [36]	To fill knowledge gaps on the development of adaptive postural control in infants born preterm	Cohort Study (Prospective Observational Study)	0% fullterm	Conformat Pressure-Sensitive Mat	Infants born preterm did not alter the postural variability in the caudal–cephalic direction in response to a visual stimulus prior to 4 months of age. They were able to adapt postural variability in the medial–lateral direction at 2.5 months of age.
Dusing et al., 2014 [7]	To investigate group differences in postural variability between infants born preterm and at risk for developmental delays or disability and infants born full term with typical development, during the emergence of early behaviors	Cross-Sectional Study	55% fullterm	Conformat Pressure-Sensitive Mat	Measures of early postural complexity are helpful in the development of interventions during the first months of life to prevent the delay in postural control strategies in preterm infants.
Dusing et al., 2014 [39]	To describe how changes in postural control during development may relate to action and perception in 3 infants born preterm with brain injury	Case Study	0% fullterm	Conformat Pressure-Sensitive Mat	Excessive postural complexity and reduced postural complexity alter the infants’ abilities to act on the world around them and use perceptual information to modify their actions.
Fallang et al., 2003 [34]	To discuss the clinical and neurophysiological data of postural behavior	Cohort Study (Prospective Observational Study)	25% fullterm	Force plate	Preterm infants show a relatively immobile postural behavior and maximum velocity of COP was substantially lower than full-term infants.
Fallang et al., 2005 [35]	To investigate whether parameters of nonoptimal reaching and reduced COP behavior at an early age are associated with dysfunctional neuromotor and behavioral development at school age.	Cohort Study (Retrospective Observational Study)	19% fullterm	Force plate	In preterm infants who do not develop CP, a lack of successful reaching at 4 months and an inadequate quality of reaching at 6 months (corrected age) are sensitive markers of clinically significant forms of brain dysfunction.
Kniaziew-Gomoluch et al., 2023 [36]	To assess reliability and validity of force plates to measure posture in preterm infants	Cohort Study (Prospective Observational Study)	0% fullterm	Force plate	Comparative analysis between the groups of infants with normal FMs and abnormal FMs in supine showed significant differences for all parameters that described spontaneous COP displacement.
Prosser et al., 2022 [24]	To investigate the ability of biomechanical measures of early postural control to distinguish infants with future impairment in motor control.	Cohort Study (Prospective Observational Study)	53% fullterm	Play and Neuro-Developmental Assessment (PANDA) gym	Quantitative methods of measuring postural control in infants born preterm and who are still hospitalized are feasible and show promise for early detection of motor impairment.

Key: COP = center of pressure; FM = Fidgety Movements.

**Table 2 diagnostics-13-03473-t002:** Quality assessment utilizing the ROB2: revised Cochrane risk-of bias tool.

ROB2 Quality Ratings									
Areas of Quality Assessed	Fallang et al., 2003 [33]	Fallang et al., 2005 [34]	Dusing et al., 2005 [38]	Dusing et al., 2009 [37]	Dusing et al., 2014 [7]	Dusing et al., 2014 [39]	Dusing et al., 2016 [36]	Prosser et al., 2022 [24]	Kniaziew-Gomoluch et al., 2023 [36]
Domain 1 Risk-of-bias-judgement: Risk of bias arising from the randomization process	High Concern	High Concern	High Concern	High Concern	High Concern	High Concern	High Concern	High Concern	High Concern
Domain 2 Risk-of-bias-judgement: Risk of bias due to deviations from the intended interventions (effect of assignment to intervention)	High Concern	High Concern	Low Concern	Low Concern	Low Concern	Low Concern	Some Concern	Low Concern	Some Concern
Domain 3 Risk-of-bias-judgement: Missing outcome data	High Concern	Some Concern	Some Concern	Low Concern	Low Concern	Low Concern	Some Concern	Some Concern	Low Concern
Domain 4 Risk-of-bias-judgement: Risk of bias in measurement of the outcome	Low Concern	Some Concern	Low Concern	Low Concern	Low Concern	Low Concern	Low Concern	Low Concern	Low Concern
Domain 5 Risk-of-bias-judgement: Risk of bias in selection of the reported result	High Concern	Some Concern	Low Concern	Low Concern	Low Concern	Low Concern	Low Concern	Some Concern	Low Concern
Overall Risk of Bias	High Concern	High Concern	High Concern	High Concern	High Concern	High Concern	High Concern	High Concern	High Concern

## Data Availability

Data are contained within the article.

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
