# Peer review of "Postural Control Measurements to Predict Future Motor Impairment in Preterm Infants: A Systematic Review"

_diagnostics, 2023, doi:10.3390/diagnostics13223473_

Round 1
Reviewer 1 Report
Comments and Suggestions for Authors
This is a well-written, very interesting systematic review. My comments are:
1.line 35, “which”. Please explain…
2.lines 43-44, “neonatal growth and development. I believe you refer to neonatal neuromotor growth and development. If you don’t, people working in NICU’s could be offended..
3.line 63, “above”. This if the first time extreme prematurity is mentioned.
4.line 147. Please correct “resulted”
5.line 305. Could you, please, comment on how effectively can these devices be disinfected as the leading cause of death in NICU is infection?
Author Response
Reviewer 1
Comments and Suggestions for Authors
This is a well-written, very interesting systematic review. My comments are:
1.line 35, “which”. Please explain…
- This sentence was rephrased to clarify as follows: Preterm infants are also more likely to display abnormal or absent fidgety movements, [5,6] which is a highly sensitive indicator of future motor impairment at 12 weeks of age.
2.lines 43-44, “neonatal growth and development. I believe you refer to neonatal neuromotor growth and development. If you don’t, people working in NICU’s could be offended..
- Thank you, this sentence was revised as follows: The neonatal intensive care unit (NICU) environment is not optimal for neonatal neuromotor development for a variety of reasons including noise levels, lighting, and other noxious stimuli, as well as suboptimal musculoskeletal support [9].
3.line 63, “above”. This if the first time extreme prematurity is mentioned.
- Thank you for noting this. The introductory phrasing was removed.
4.line 147. Please correct “resulted”
- This correction was made.
5.line 305. Could you, please, comment on how effectively can these devices be disinfected as the leading cause of death in NICU is infection?
- Thank you. At reviewer’s suggestion, we added the following to the discussion section (lines 363-367): While most studies did not address if force plate and pressure mat technologies can be easily disinfected between uses in the highly vulnerable preterm infant population, the carbon fiber core dragon plate used in the PANDA gym can be easily cleaned with soap and water or disinfectant wipes between use. This plate is covered with foam padding or a blanket for infant comfort which would be used only once for each infant to decrease the spread of infection.
Reviewer 2 Report
Comments and Suggestions for Authors
The paper presented a systematic review of postural control measurements to predict future motor impairments in preterm infants. This is a well-written manuscript that only needs to undergo a few changes which are listed below.
- As both linear and non-linear measures of postural control were used in studies included in this systematic review, the authors are suggested to add a brief explanation of these measures and their use in the assessments of different aspects of postural control (see Effects of cognitive load on the amount and temporal structure of postural sway variability in stroke survivors. Experimental Brain Research. 2018 Jan;236:285-96.)
- Page 1, lines 34-36: The sentence "Preterm infants are also more likely to …" is grammatically incorrect and should be re-written.
- Page 2, line 42: The sentence "and contributes to the difficulty these infants have maintaining midline orientation" should be rewritten.
- Page 2, line 56: "range from" should be "ranges from"
- Page 2, line 65: " Neonatal Intensive Care Unit (NICU)" should be "NICU"
- Page 3, line 74: "validity: but few standardized assessments …" should be "validity. Few standardized assessments "
- Page 4: Please check Figure 1. It seems that a mistake has occurred in the box entitled "17 full-text articles excluded". For example, It has been mentioned that 13 articles were excluded due to not being clinical trials while "being the clinical trial" has not been mentioned as an inclusion criterion. The same condition is observed for massage and parent outcomes.
- Page 7, the third row of the table: "4 mo" and "6 mo" should be "4 months" and "6 months".
- Page 10-11 and throughout the text: "Path Length" should be "path length"
- Page 10, line 235: "maximum velocity" should be "maximum velocity (Vmax)"
- Page 10, line 252: Change the "test re-test data" to "test-retest data"
- Page 11, line 277: "are" should be removed.
- Page 12, line 324: "which physical therapy interventions" should be changed to "which occupational therapy and physical therapy interventions"
- Page 12, lines 319-321: The sentence "Studies presented in this systematic review support the association between reduced postural complexity before development of midline head control as an indicator of future motor delay" is grammatically incorrect and should be revised. Further, because of the importance of finding an indicator of future motor delay in infants, the authors are suggested to add the revised version of this sentence to the abstract.
- Page 12, line 334 and Page 1, line 21: "severity" should be "impairment"
Comments on the Quality of English Language
Minor editing of English language is needed.
Author Response
Reviewer 2
Comments and Suggestions for Authors
The paper presented a systematic review of postural control measurements to predict future motor impairments in preterm infants. This is a well-written manuscript that only needs to undergo a few changes which are listed below.
- As both linear and non-linear measures of postural control were used in studies included in this systematic review, the authors are suggested to add a brief explanation of these measures and their use in the assessments of different aspects of postural control (see Effects of cognitive load on the amount and temporal structure of postural sway variability in stroke survivors. Experimental Brain Research. 2018 Jan;236:285-96.)
- Thank you for this suggestion and source. We have now added additional language in the results section to better define these measures as follows (lines 213-225): Both linear and non-linear measures of postural control were used in the reviewed studies. The most commonly used metric, COP, is considered to be a linear measure, as it reflects overall postural control by quantifying the variability of COP. Generally in the adult population, high variability in COP is interpreted as postural instability, whereas lower COP variability indicates greater postural control; however, the reviewed studies mostly pointed to greater COP variability in healthy fullterm control infants as compared to preterm infants, who demonstrated less complexity in movement [36,37].
Non-linear metrics, which incorporate time into COP variability, were also used for postural control analysis. These non-linear metrics quantify the amount of randomness, fluctuation, and unpredictability during dynamic movement. As observed in linear postural metrics, preterm infants actually demonstrated small amounts of entropy, or randomness, than fullterm infants, indicating less variability of complex movement [24].
Page 1, lines 34-36: The sentence "Preterm infants are also more likely to …" is grammatically incorrect and should be re-written.
- Thank you. This sentence was edited as follows (lines 36-40): Preterm infants demonstrate lower quality of spontaneous movements, with descriptions such as low fluency, less variety, and impaired sequencing [4]. Preterm infants are also more likely to display abnormal or absent fidgety movements, [5,6] which is a highly sensitive indicator of future motor impairment at 12 weeks of age.
Page 2, line 42: The sentence "and contributes to the difficulty these infants have maintaining midline orientation" should be rewritten.
- This sentence has been edited as follows (lines 44-46): . Increased muscle power [8], results in the hyperextended posture commonly observed in preterm infants. This posture further leads to difficulties maintaining midline orientation [8].
Page 2, line 56: "range from" should be "ranges from"
- This edit has been made.
Page 2, line 65: " Neonatal Intensive Care Unit (NICU)" should be "NICU"
- This edit has been made.
Page 3, line 74: "validity: but few standardized assessments …" should be "validity. Few standardized assessments "
- This edit has been made.
Page 4: Please check Figure 1. It seems that a mistake has occurred in the box entitled "17 full-text articles excluded". For example, It has been mentioned that 13 articles were excluded due to not being clinical trials while "being the clinical trial" has not been mentioned as an inclusion criterion. The same condition is observed for massage and parent outcomes.
- Thank you for finding this discrepancy. Figure 1 has been corrected.
Page 7, the third row of the table: "4 mo" and "6 mo" should be "4 months" and "6 months".
- This correction has been made
Page 10-11 and throughout the text: "Path Length" should be "path length"
- This edit has been made throughout
Page 10, line 235: "maximum velocity" should be "maximum velocity (Vmax)"
- This edit was made
Page 10, line 252: Change the "test re-test data" to "test-retest data"
- This correction was made.
Page 11, line 277: "are" should be removed.
- This edit was made.
Page 12, line 324: "which physical therapy interventions" should be changed to "which occupational therapy and physical therapy interventions"
- This edit was made.
Page 12, lines 319-321: The sentence "Studies presented in this systematic review support the association between reduced postural complexity before development of midline head control as an indicator of future motor delay" is grammatically incorrect and should be revised. Further, because of the importance of finding an indicator of future motor delay in infants, the authors are suggested to add the revised version of this sentence to the abstract.
Page 12, line 334 and Page 1, line 21: "severity" should be "impairment"
- This edit was made.
Reviewer 3 Report
Comments and Suggestions for Authors
This is a literature review on Postural Control Measurements to Predict Future Motor Impairment in Preterm Infants. Although the study focuses on most commonly used tools, I believe, they should provide some more insights related to wearable technologies, if they can be any use for infants, as in this study Simultaneous validation of wearable motion capture system for lower body applications: over single plane range of motion (ROM) and gait activities" Biomedical Engineering / Biomedizinische Technik, vol. 67, no. 3, 2022, pp. 185-199. https://doi.org/10.1515/bmt-2021-0429
Any use of AI based predictive algorithms and more futuristic use of the common techniques could be highlighted since review papers should provide some futuristic vision for the researchers.
Also, there is no images in the paper, this makes it a bit dry and difficult to understand the explained products without any schematics. A flow chart algorithm could be used to define the steps in the process to provide a better explanation.
Moreover, you could provide more specific detail ( hip, spine, knee, shoulder joints etc) about the posture control, referring to the joint for each system as in the following paper Wearable Motion Capture System Evaluation for Biomechanical Studies for Hip Joints." ASME. J Biomech Eng. April 2021; 143(4): 044504. https://doi.org/10.1115/1.4049199
Comments on the Quality of English LanguageEnglish language quality is good in overall and minor errors could be corrected.
Author Response
Reviewer 3
Comments and Suggestions for Authors
This is a literature review on Postural Control Measurements to Predict Future Motor Impairment in Preterm Infants. Although the study focuses on most commonly used tools, I believe, they should provide some more insights related to wearable technologies, if they can be any use for infants, as in this study Simultaneous validation of wearable motion capture system for lower body applications: over single plane range of motion (ROM) and gait activities" Biomedical Engineering / Biomedizinische Technik, vol. 67, no. 3, 2022, pp. 185-199. https://doi.org/10.1515/bmt-2021-0429
Any use of AI based predictive algorithms and more futuristic use of the common techniques could be highlighted since review papers should provide some futuristic vision for the researchers.
- Two paragraphs were added to the introduction to outline the increased potential for use of technology in early motor impairment detection, including the use of wearable sensors. The paragraph also addresses use of AI to establish algorithms that can aid in early detection of delays (lines 99-11): Pediatric therapist researchers are advocating for expanded use of technology in clinical settings to detect early motor delay. Postural measurement tools, including portable pressure-sensitive mats and force plates, may be effective in measuring COP in preterm infants, and therefore useful for detecting early delays in high-risk infant populations [24]. Additionally, the use of wearable sensors including inclusive clothing, exoskeletons, and smart tracking devices, are being examined in high-risk infant populations to determine potential ways that this technology can assist with understanding how specific movement characteristics may enhance or detract from the infant’s developmental trajectory.
- The use of such technologies expands opportunities for the use of artificial intelligence to assist in the early and accurate diagnosis of neuromotor disabilities. Gaining a better understanding of the role of early postural control deviations as measured by force plates or sensors allows researchers to use these technologies to build algorithms that detect and quantify movements associated with future motor impairments.
Also, there is no images in the paper, this makes it a bit dry and difficult to understand the explained products without any schematics. A flow chart algorithm could be used to define the steps in the process to provide a better explanation.
- The authors would be happy to seek permissions for use of photos from the reviewed articles with the appropriate paperwork from Diagnostics. Please advise how to move forward with this.
Moreover, you could provide more specific detail ( hip, spine, knee, shoulder joints etc) about the posture control, referring to the joint for each system as in the following paper Wearable Motion Capture System Evaluation for Biomechanical Studies for Hip Joints." ASME. J Biomech Eng. April 2021; 143(4): 044504. https://doi.org/10.1115/1.4049199
- Thank you for this suggestion. Because this paper is limited to force plate and pressure mat applications, and not wearable technologies, the authors believe these further explanations may go beyond the scope of the originally intended manuscript.
Round 2
Reviewer 2 Report
Comments and Suggestions for Authors
Dear Authors,
I appreciate your efforts to address the comments and suggestions that I made in my previous review. I think you have improved the quality and clarity of your paper significantly.
However, before your paper can be accepted for publication, there are still some minor grammatical errors that need to be corrected which are listed below.
- Page 2, line 44: "difficulties maintaining" should be "difficulties in maintaining"
- Page 10, line 200: "The most commonly used metric, COP, is considered …. the variability of COP [40]" is suggested to be re-written as follows:
"Linear measures such as path length quantify the amount of COP variability [40]."
- Page 14, line 336: "While" should be removed.
Page 14, line 336: "if" should be "whether"
- Page 14, lines 337-338: "preterm infant population, the carbon fiber core" is suggested to be "preterm infant population. However, the carbon fiber core "
Comments on the Quality of English LanguageMinor English language editing is needed.
Author Response
Dear Reviewer 2,
Thank you very much for your additional feedback. I have gone through line by line and made all edits as suggested.